# Thermodynamics Investigation and Artificial Neural Network Prediction of Energy, Exergy, and Hydrogen Production from a Solar Thermochemical Plant Using a Polymer Membrane Electrolyzer

**DOI:** 10.3390/molecules28062649

**Published:** 2023-03-14

**Authors:** Atef El Jery, Hayder Mahmood Salman, Rusul Mohammed Al-Khafaji, Maadh Fawzi Nassar, Mika Sillanpää

**Affiliations:** 1Department of Chemical Engineering, College of Engineering, King Khalid University, Abha 61421, Saudi Arabia; 2National Engineering School of Gabes, Gabes University, Ibn El Khattab Street, Gabes 6029, Tunisia; 3Department of Computer Science, Al-Turath University College Al Mansour, Baghdad 10013, Iraq; 4Building and Construction Techniques Engineering Department, Al-Mustaqbal University College, Babylon 11702, Iraq; 5Integrated Chemical Biophysics Research, Faculty of Science, University Putra Malaysia (UPM), Serdang 43400, Selangor, Malaysia; 6Department of Chemistry, Faculty of Science, University Putra Malaysia (UPM), Serdang 43400, Selangor, Malaysia; 7Faculty of Science and Technology, School of Applied Physics, University Kebangsaan Malaysia, Bangi 43600, Selangor, Malaysia; 8International Research Centre of Nanotechnology for Himalayan Sustainability (IRCNHS), Shoolini University, Solan 173212, Himachal Pradesh, India

**Keywords:** polymer electrolyte membrane, energy efficiency, current density, concentrated solar system, exergy efficiency, machine learning, artificial neural network

## Abstract

Hydrogen production using polymer membrane electrolyzers is an effective and valuable way of generating an environmentally friendly energy source. Hydrogen and oxygen generated by electrolyzers can power drone fuel cells. The thermodynamic analysis of polymer membrane electrolyzers to identify key losses and optimize their performance is fundamental and necessary. In this article, the process of the electrolysis of water by a polymer membrane electrolyzer in combination with a concentrated solar system in order to generate power and hydrogen was studied, and the effect of radiation intensity, current density, and other functional variables on the hydrogen production was investigated. It was shown that with an increasing current density, the voltage generation of the electrolyzer increased, and the energy efficiency and exergy of the electrolyzer decreased. Additionally, as the temperature rose, the pressure dropped, the thickness of the Nafion membrane increased, the voltage decreased, and the electrolyzer performed better. By increasing the intensity of the incoming radiation from 125 $W/m^{2}$ to 320 $W/m^{2}$, the hydrogen production increased by 111%, and the energy efficiency and exergy of the electrolyzer both decreased by 14% due to the higher ratio of input electric current to output hydrogen. Finally, machine-learning-based predictions were conducted to forecast the energy efficiency, exergy efficiency, voltage, and hydrogen production rate in different scenarios. The results proved to be very accurate compared to the analytical results. Hyperparameter tuning was utilized to adjust the model parameters, and the models’ results showed an MAE lower than 1.98% and an $R^{2}$ higher than 0.98.

## 1. Introduction

Unlike the fossil fuels available in nature, hydrogen does not have a natural source and must be produced by human-made methods [1]. Many renewable energy sources can be stored, through which hydrogen fuel can be produced. These sources can be solar energy, wind, and biomass. The method of hydrogen production introduced by Dincer [2] has been extensively researched. Other researchers have also investigated methods for hydrogen production from energy sources [3,4,5,6,7]. Hydrogen production is best carried out with water when using sustainable sources on a large scale [8]. Due to its renewable, safe, and clean nature, solar energy is considered one of the best options. As a source of thermal energy for the high-temperature process, Steinfeld [9] investigated the thermochemical production of hydrogen using solar radiation. In his study, he demonstrated that the thermochemical production of hydrogen by the electrolysis of water using electricity generated by the sun competes with conventional fossil fuel methods, since electrolysis reduces carbon dioxide production and pollution. The chemical decomposition of water, thermochemical cycles or the reforming of hydrocarbons, and electrolysis require electricity to decompose water. Many scientists are interested in combining the water electrolysis process with a renewable energy system that provides power and heat for hydrogen production [10]. The storage of oxygen and hydrogen in space for an extended period of time is impractical. Just as scientists have determined the climatic parameters needed by the body to sustain life on earth, aerospace scientists have similarly estimated the climatic parameters needed to sustain life in space. On earth, plants play an important role in removing carbon dioxide and producing oxygen, but in space, other methods are used to remove carbon dioxide and produce water and oxygen. In space shuttles, fuel cells produce electricity by combining hydrogen and oxygen. In a fuel cell, electricity is generated by a chemical reaction similar to that of a battery. However, the fuel is continuously supplied by oxygen and hydrogen. As a consequence of this reaction, water is produced, which can be used to make oxygen for breathing and hydrogen to fuel the electrolyzer. Every electrolyzer cell contains polar plates, an anode, and a cathode, as well as electrolytes and catalysts in the gas diffusion layer [11]. Electrolyzer technology is divided into four general categories: alkaline electrolyzers, whose electrolyte is a liquid (such as potassium hydroxide); proton and anion exchange membrane electrolyzers, both of which have a solid polymer electrolyte; and finally, solid oxide electrolyzers, which have ceramic membranes. Polymer membrane electrolyzers have attracted most attention due to their various advantages, including their lack of toxic material production, better systematic management, pure hydrogen generation, higher efficiency, lack of pollution generation, and higher current density [12,13,14]. A polymer membrane electrolyzer consists of three parts [15]:At a local low current density, when the effects of ohmic losses can be ignored, the activation voltage drop obtained from the Volmer–Butler equation is the dominant drop.In the middle of the diagram, it is linear, and ohmic drops cause the voltage drop.The dominant drop is the concentration voltage drop at a high current density.

Several researchers have modeled polymer membrane electrolyzers. Koka et al. [16] investigated the impact of multi-material anode catalysts on the performance of polymer membrane electrolyzers and the effect of the catalytic layer on various types of membranes. As a steady-state and variant-temperature model, Harrison et al. [17] used a semiempirical zero-dimension model. Based on experimental results and a non-linear curve to relate current to voltage, the authors characterized 20 cells using a polymer membrane electrolyzer stack. Furthermore, Dale et al. [18] developed a semiempirical model of a polymer membrane electrolyzer based on thermodynamic laws to determine the characteristics of the electrolyzer. After conducting experimental analyses on commercial polymer membrane electrolyzers, Santarelli et al. used regression to analyze the results. A high-pressure polymer membrane electrolyzer with various levels of stack flow was investigated in [19] to determine the effect of different working parameters on the supply voltage. An experimental study by Chandesris et al. [20] investigated the effects of current density and temperature on the degradation of a membrane using a one-dimensional numerical model. As a result of their research, the authors discovered that the greatest amount of membrane destruction occurred within the container, and that the temperature had a strong influence on this process.

Other studies have investigated hydrogen production systems using photovoltaic cells and polymer membrane electrolyzers [21,22,23,24]. Scamman et al. [25] investigated the combination of solar energy systems using photovoltaic cells and polymer membrane electrolyzers in three weather conditions. Gibson and Kelley [26] investigated the combined system of photovoltaic cells and a polymer membrane electrolyzer by adjusting the voltage and maximum power. They optimized the photovoltaic output to the working voltage of the polymer membrane electrolyzer and photovoltaic cells. This optimization process raised the hydrogen production efficiency by up to 12% for this system, which provided enough hydrogen for the fuel cell used in the vehicle. Paul and Andris [27] investigated a hydrogen renewable energy system to generate power in remote areas. The preliminary experimental results showed that about 95% of the energy was transformed compared to the theoretical state. The system was also fully investigated and justified from an economic point of view.

Although there has been much research on polymer membrane electrolyzers [28,29,30,31,32,33], none of the research in this area has involved the energy and exergy analysis of a polymer membrane electrolyzer system in combination with a solar concentrating system. This study investigated hydrogen production using a polymer membrane electrolyzer system that provides power from a concentrated solar system. Hence, it is important to introduce the system first and then analyze its energy and efficiency by stating the conditions within the system that govern its operation. Additionally, only a few studies have combined their results with the multi-disciplinary field of machine learning to propose predictive models. In this study, we used artificial neural networks to predict four important parameters in the hydrogen production process, and all models were studied thoroughly to attain the best results.

## 2. System Description

Figure 1 presents a schematic view of the investigated system. This system consisted of four parts: a tower, a solar thermal engine, a generator, and a polymer membrane electrolyzer. The heliostats and solar tower subsystem consisted of many mirrors arranged in a circular pattern to reflect solar radiation to a central receiver atop the solar tower. It has been demonstrated that solar tower technology is effective even at temperatures as high as 2200 K [34]. As a result of this feature, this technology is suitable for use in conjunction with other high-temperature systems. The assumptions and numerical values for this part of the system are presented in Table 1. When the solar radiation is reflected, it is focused on the central receiver and transferred to the heat engine. The engine consumes some radiation and turns it into mechanical work. Then, the mechanical work is sent to the electric generator to produce electricity and power the electrolyzer.

The membrane electrolyzer transfers polymer molecules containing water and protons and decomposes them into oxygen, protons, and electrons. The electrons leave the cell from an external circuit, combine with protons in the cathode, and release hydrogen [11]. The reactions in the cathode and anode of the polymer membrane electrolyzer are as follows:
The anode$H_{2}O\to \;2H^{+}+\frac{1}{2}O_{2}+2e^{-}$
The cathode
$2H^{+}+2e^{-}\to H_{2}$The whole reaction $H_{2}O_{\left(l\right)}\to H_{2}{}_{\left(g\right)}+\frac{1}{2}O_{2}{}_{\left(g\right)}$

## 3. Thermodynamic Analysis

The thermodynamic analysis of the cycle was carried out using Engineering Equation Solver (EES). When solving the problem, we took into account the following assumptions. Firstly, potential and kinetic energy changes were not taken into account. The system maintained a steady state. Chemical exergy was considered only for polymer membrane electrolyzers.

### 3.1. Solar Tower

Using Equation (1), we could determine the total solar radiation reaching the heliostat [35].
(1)$$Q_{h}=A_{h}\times I$$
where $A_{h}$ is the heliostat field area in $m^{2}$, and *I* is the inlet radiation intensity in $W/m^{2}$. In terms of the optical efficiency of the heliostat field, the rate of heat reflection to the receiver was defined as follows:(2)$$\eta_{opt}=\frac{Q_{rec}}{Q_{h}}$$
where the subscripts rec and h denote the receiver and heliostat. The efficiency of the receiver could also be determined by comparing the rate of heat absorbed to the rate of heat reflected.
(3)$$\eta_{rec}=\frac{Q_{abs}}{Q_{rec}}$$

In Equation (3), $Q_{abs}$ is the absorbed heat. The exergy rate in relation to the heat flux was defined as follows [35].
(4)$$Ex_{h}=Q_{h}\left(1-\frac{T_{0}}{T_{sun}}\right)$$
where $T_{0}$ and $T_{sun}$ are the standard and solar temperatures. As a consequence, the reflected radiation’s exergy was expressed as:(5)$$Ex_{rec}=Q_{rec}\left(1-\frac{T_{0}}{T_{rec}}\right)$$

$T_{rec}$ is the receiver temperature. The following equation defines the specific exergy of the fluid in each state [35].
(6)$$ex_{i}=C_{P}{}_{i}\left(T_{i}-T_{0}\right)-T_{0}\left[C_{P}{}_{i}\ln\left(\frac{T_{i}}{T_{0}}\right)\right]$$
where $C_{p}$ is the specific heat. As a result of the absorption of heat, an exergy rate was generated.
(7)$$Ex_{abs}=Q_{abs}\left(1-\frac{T_{0}}{T_{rec}}\right)$$

The solar towers had the following energy and energy efficiency.
(8)$$\eta_{solar}=\frac{Q_{abs}}{Q_{h}}$$
(9)$$\psi_{solar}=\frac{{\mathrm{Ex}}_{abs}}{{\mathrm{Ex}}_{h}}$$

### 3.2. Heat Engine

Mechanical work was defined using the following equations [36].
(10)$$\dot{W}=\eta_{HE}{\dot{Q}}_{abs}$$
(11)$$\eta_{HE}=1-\frac{T_{0}}{T_{\mathrm{rec}}}$$

In these equations, $\eta_{HE}$ is the ideal efficiency of the Carnot cycle combustion engine, corresponding to the maximum energy received for each heat engine. A heat engine has a lower energy efficiency than the Carnot cycle, which is around 30 to 50%; thus, its efficiency is less than the Carnot cycle. In this article, this value was considered 45% [36].

### 3.3. Electric Generator

As defined in [36], electric generators have a high energy efficiency.
(12)$$\eta_{GEN}=\frac{P_{E}}{P_{M}}$$

In this equation, $P_{E}$ is the electrical work, and $P_{M}$ is the mechanical work. Assuming that the electrical output and mechanical input remain constant, a generator’s energy efficiency is similar to that of an electric motor. In an electric generator, the losses include the core’s iron losses, the armature’s copper losses, and the voltage drop in the diode bridge, usually 50 to 62% [36]. The core consists of hysteresis losses and losses caused by eddy currents in the armature core, and copper losses occur due to the current passing through the armature excitation coils and other coils in the generator. The final expression for the exergy efficiency of the electric generator is presented below.
(13)$$\psi_{GEN}=\frac{P_{E}}{P_{M}}=\frac{VI}{\tau \times \omega }=\frac{VI}{\tau \times \frac{2\pi rpm}{60}}$$

In this equation, τ, ω, *V*, and I are the torque, revolutions per minute, voltage, and current, respectively.

### 3.4. Polymer Membrane Electrolyzer

The fixed parameters used in modeling the polymer membrane electrolyzer are presented in Table 2 [37]. A polymer membrane electrolyzer exhibits a reversible voltage based on the Gibbs free energy of the reaction. This voltage corresponds to the ideal electrolyzer cell in reversible and isothermal conditions.

All the electrical energy obtained in an external circuit is available in fuel cells. The concept for the electrolyzer is that all the driving force produced is available for hydrogen production without any loss. The remainder of the voltage drops increase the reversible voltage. The effects of the concentration were considered in terms of the Nernst equation and the concentration drops, which were omitted in this article due to their small number, and only ohmic drops and activation were considered. The Gibbs free energy is defined as shown in Equation (14).
(14)ΔG=ΔH−TΔS
where ΔH is the enthalpy change in J/mol, and ΔS is the entropy change. Under ideal conditions, the heat produced during the reaction is continuously removed from the system, so the system’s temperature stays constant. The amount of removed heat equals the entropy change multiplied by the working temperature. Equation (15) was used for calculating the reversible potential [38]. The open circuit voltage for the polymer membrane electrolyzer was obtained using the Nernst equation [39].
(15)$$V_{0}=-\frac{\Delta G}{2F}$$
(16)$$V_{0}=1.229-0.9\times 10^{-3}\left(T-298\right)+\frac{RT}{nF}\ln\left(\frac{P_{H_{2}}P_{O_{2}}^{0.5}}{P_{H_{2}O}}\right)$$

In the above relation, T is the electrolyzer temperature; *R* is the global gas constant, which is equal to 8.314 J/Kmol; n is the number of exchanged electrons; and F is the Faraday constant, which is equal to 96,485 C/mol. The single-cell voltage of the electrolyzer was defined as shown in Equation (17). In Equation (17), $V_{act}$ is the activation voltage drop in the anode and cathode, and $V_{ohm}$ is the important voltage drop in volts.
(17)$$V=V_{0}+V_{act}+V_{ohm}$$

#### 3.4.1. Activation Voltage Drop

An increase in the activation voltage indicates that the electrons are ready to participate in the electrochemical reaction. During the chemical process, some of the voltage applied to the electrolyzer is lost as electrons are transferred from the surface of the electrodes to the electrodes. Using the Volmer–Butler equation, the activation energy was modeled on both the cathode and anode sides as a function of the activation voltage drop.
(18)$$V_{act,i}=\frac{RT}{F}\sinh^{-1}\left(\frac{J}{2J_{0,i}}\right)$$
(19)$$J_{0,i}=J_{i}^{ref}EXP\left(-\frac{E_{act,i}}{RT}\right)\;\;\;\;\;\;\;\;i=a,c$$
where *R* is the universal gas constant, *T* is the temperature, *F* is the Faraday constant, *J* is the current density, and $E_{act}$ is the activation energy. The material type and electrode porosity; the concentration, size, and distribution of catalyst particles on the electrodes; and the working temperature determine the current exchange density. Regarding platinum catalysts, electrodes with anodes have current exchange densities ranging from $10^{-8}$ to $10^{-5}$, and electrodes with cathodes have current exchange densities ranging from 10 to $10^{-1}$ $A/m^{2}$ [13]. According to this article, the current exchange density for an anode based on a platinum catalyst is 10 $A/m^{2}$, and the current exchange density for a cathode is 10 $A/m^{2}$ [40].

#### 3.4.2. Activation Voltage Drop

As a result of the flow of electrons, an ohmic voltage drop is created, which equates to the resistance created by the polymer membrane electrolyzer against the flow of electrons. It is important to note that the voltage drop is dependent upon the type of polymer membrane electrolyzer and the type of electrodes. The current density ($A/{\mathrm{cm}}^{2}$) is linearly related to the ohmic voltage drop. The ohmic voltage drop is the resistance of the electrolyzer against the transfer of protons and is defined as follows [11].
(20)$$V_{ohm}=\frac{L}{\sigma_{mem}}J$$
(21)$$\sigma_{mem}=\left(0.005139\lambda -0.00326\right)EXP\left[1263\left(\frac{1}{303}-\frac{1}{T}\right)\right]$$
where $\sigma_{mem}$ is the ionic conductivity of the Nafion membrane in terms of 1/Ωcm [41]. Due to the fact that proton transfer occurs on the surface of the membrane using water molecules, the water content is a significant factor in determining the membrane’s ionic conductivity. λ is the water content, which is very important to determine due to the humidity changes of the membrane in fuel cells. In fuel cells, due to the abundance of water on the anode and cathode (due to the transfer phenomenon), the entire membrane can be considered humid. λ is usually in the range of 14 to 21; in this research, its value was considered to be 21 [13]. In order to determine the performance of the system, efficiencies were utilized.
(22)$$\eta_{pem}=\frac{{\dot{N}}_{H_{2},out}\times LHV_{H_{2}}}{W_{elec}+Q_{heat,pem}}$$
(23)$$\psi_{pem}=\frac{{\dot{N}}_{H_{2,out}}\times Ex_{H_{2}}}{W_{elec}+Ex_{heat,pem}}$$

The amount of hydrogen, oxygen, and water output from the electrolyzer was obtained from Equations (24)–(26) [37].
(24)$${\dot{N}}_{H_{2,out}}=\frac{J}{2F}={\dot{N}}_{H_{2}O,reacted}$$
(25)$${\dot{N}}_{O_{2,out}}=\frac{J}{4F}$$
(26)$${\dot{N}}_{H_{2}O,\mathrm{out}}={\dot{N}}_{H_{2}O,\mathrm{in}}-{\dot{N}}_{H_{2}O,\mathrm{reacted}}={\dot{N}}_{H_{2}O,\mathrm{in}}-\frac{J}{2F}$$
(27)$$W_{elec}=JV$$

The water flow rate to the electrolyzer was 0.005 kg/s, and the hydrogen exergy was defined as follows:(28)$$Ex_{H_{2}}=Ex_{H_{2}}^{ph}+Ex_{H_{2}}^{ch}$$
where $Ex_{H_{2}}^{ph}$ is the physical exergy of hydrogen obtained from Equation (6), and $Ex_{H_{2}}^{ch}$ is the chemical exergy of hydrogen, whose value is 236.09 kJ/kmol [42]. Irreversibility occurs in the polymer membrane electrolyzer due to entropy generation. This parameter was calculated as follows [36].
(29)$$\sigma =2F\left(V_{act,a}+V_{act,c}+V_{ohm}\right)$$

If σ≥TΔS, the heat produced due to the irreversibility of the system is equal to or greater than the heat required for decomposition, and no external heat is required for the polymer membrane electrolyzer. Therefore, $Q_{heat,pem}=Ex_{heat,pem}=0$. Notably, the excess produced heat was assumed to be released into the environment through radiation. If σ<TΔS, the produced heat is less than the required heat, and additional heat is required. The amount of heat input into the electrolyzer could be measured by Equation (30).
(30)$$Q_{heat,pem}=\left[T\Delta S-\sigma \right]N_{H_{2}O,reacted}=\left[T\Delta S-\sigma \right]\frac{J}{2F}$$
(31)$$Ex_{heat,pem}=Q_{heat,pem}\left(1-\frac{T_{0}}{T}\right)\;$$

### 3.5. Overall Efficiency

After calculating the efficiency of each part of the system, the overall energy efficiency and exergy of the entire solar hydrogen production system were obtained as follows:(32)$$\eta_{energy}=\eta_{solar}\times \eta_{HE}\times \eta_{GEN}\times \eta_{pem}$$
(33)$$\psi_{energy}=\psi_{solar}\times \psi_{HE}\times \psi_{GEN}\times \psi_{pem}$$

## 4. Results and Discussion

### 4.1. Electrolyzer Validation

In order to validate the electrolyzer model investigated in this article, our results were checked against the results of experimental tests [43]. The electrolyte used in the test was Nafion, with a thickness of 50 μm, and platinum was used for the electrode on the cathode. Figure 2 shows the polarization diagram of the electrolyzer with the working conditions listed in Table 3.

The results of this study showed good agreement with the experimental results of [43]. The ohmic voltage drop and anode and cathode activation are shown in Figure 3. The $V_{0}$ value was 1.17 in Figure 3. The ohmic voltage drop was very low and increased continuously with an increasing current density. The reason for this was that the ionic conductivity of the membrane was high at λ=21 and a working temperature of 353 K. Although the value of the capacitor activation voltage drop was higher than the ohmic voltage drop, it was much smaller than the anode activation voltage drop. This was because of the reaction’s faster kinetics on the cathode’s surface. The current density of less than 1000 $A/m^{2}$ suddenly increased, and then the current increased uniformly with the increase in density.

As is clear from Figure 3, the cathode activation voltage drop had a small value due to the high speed of the hydrogen formation reaction, and the anode activation voltage drop had a larger value due to the low speed of the oxygen formation reaction, which determined the overall voltage drop of the electrolyzer.

### 4.2. Energy Efficiency and Exergy

Figure 4 shows the energy efficiency and exergy of the polymer membrane electrolyzer at 350 K. It is well-known that the energy efficiency and exergy curves are very similar. The input energy and output energy both increased with an increasing current density. This was because the input energy and output energy both increased. As the current density of the low-current generator increased, the electrical energy required rose significantly, while the input and output thermal energies increased linearly. Since an increase in current density corresponded to an increase in the input energy (electrical and thermal energy) at a faster rate than the output energy, i.e., hydrogen production, the flow of energy efficiency dropped with the rise in density. Due to the low temperature of most polymer membrane electrolyzers, the only thermal energy required is for heating the inlet water, and the electrolyzer’s energy and exergy efficiency is entirely dependent on electricity rather than the electrolyzer’s temperature. Due to the 100% exergy efficiency of electricity, the system’s energy and exergy efficiency were close, which demonstrated this system’s competency.

### 4.3. Effect of Functional Parameters

As shown in Figure 5, augmenting the temperature led to an increase in the energy and exergy efficiency of the electrolyzer at a 5000 $\mathrm{mA}/m^{2}$ current density. A higher temperature meant more reactions in the electrodes, leading to a greater current exchange density and reducing the activation voltage drop, according to Equation (20). The electrolyzer cell voltage decreased as the temperature increased, according to Equations (20) and (21); therefore, the electrical input decreased [37].

.

A polarization diagram of a polymer membrane electrolyzer at various temperatures and operating pressures is shown in Figure 6 and Figure 7. According to Figure 5, the voltage of the cell decreased with an increasing temperature, while the open circuit voltage decreased as the temperature increased. Accordingly, the difference between the Gibbs free energy and the Nernst voltage revealed the effects of the product and reactant concentrations. Higher temperatures accelerated the electrolyzer reaction, since the electrolyzer reaction is endothermic. However, polymer membrane electrolyzers are limited in their design due to the characteristics of the membrane. When the temperature and pressure increased in the Nafion membrane, the diffusion coefficient of $H^{+}$ ions increased, which was directly related to the membrane’s conductivity. At higher temperatures, the first effect was a decrease in voltage, followed by a decrease in the power required for the electrolyzer, while the Nafion membrane’s transmission properties increased. However, as a result of the opposite movement of $H^{+}$ ions from the anode to the cathode, the cathode was under a higher pressure, causing the voltage to increase and the reaction to be problematic.

Figure 8 shows the voltage of the electrolyzer in terms of the working temperatures with a current density of 5000 $A/m^{2}$. As the temperature increased, the cell voltage decreased due to the chemical reaction’s acceleration and the increase in the current exchange density. As a result, the activation and ohmic voltage dropped.

Figure 9 illustrates the polarization diagram of the polymer membrane electrolyzer with different membranes. At a current density of less than 1000 $A/m^{2}$, the change in membrane thickness had little effect on the electrolyzer voltage, but for higher values of current density, the effect of the thickness on the electrolyzer voltage increased. With the increase in the thickness of the membrane, the voltage of the polymer membrane electrolyzer increased, which was the reason for the augmentation in the ohmic voltage. The decrease in the thickness of the membrane caused a decrease in the resistance; the amount of material used; and, therefore, the price. However, if the thickness of the membrane decreased too much, it could decrease the life of the membrane, and reactive gases could pass through it.

### 4.4. The Effect of the Intensity of Solar Radiation

Figure 10 shows the effect of I on the energy and exergy efficiency of the electrolyzer. With the augmentation in solar radiation, the efficiencies decreased. This was because an increase in solar radiation caused more thermal energy to enter the receiver of the solar tower, and, as a result, more power was provided to the electrolyzer as an input. By increasing the amount of solar radiation from 125 $W/m^{2}$ to 320 $W/m^{2}$, the exergy and energy efficiencies decreased by 13.7% and 12.6%, respectively.

Figure 11 shows the effect of I on the hydrogen production and the energy efficiency of the whole system. By increasing the intensity of solar radiation from 125 $W/m^{2}$ to 320 $W/m^{2}$, the hydrogen production increased by 111%.

The generated hydrogen was directly related to the power supply of the electrolyzer. When the input power to the electrolyzer rose, the hydrogen production also increased. By increasing I from 125 $W/m^{2}$ to 320 $W/m^{2}$, the energy efficiency of the entire system decreased by 14%, which was the reason for the decrease in the efficiency of the electrolyzer.

## 5. Artificial Neural Network

Using machine-learning-based algorithms to propose predictive models has recently become common in engineering problems [44,45,46]. This is mainly because the capability of these models in prediction is spectacular [47]. Among all of these models, the artificial neural network has shown the best results and is the subject of abundant research [48].

In the current research, we used the data from the analyses to propose predictive models for energy and exergy efficiency, voltage, and $H_{2}$ production. The input parameters selected for the models were $P_{cathode},\;I,\;L,\;J,\;\mathrm{and}\;T$. The data in each output parameter were split in a 70–30% ratio, i.e., 70% of the data were employed for training, and 30% were used for testing the models. In order to better optimize the models, we used hyperparameter tuning for all the models. This procedure investigated all the available settings adjustments and selected the best model for the prediction of the output parameter. In the following, an example of this process is presented for voltage.

One of the most significant parameters of an artificial neural network is the structure of its hidden layers; thus, in Table 4, the effect of this parameter is studied. It was observed that, generally, by increasing the number of layers, the accuracy improved and then stopped at a certain point. This was basically overfitting, where the model was unable to generalize its predictions on other data and was fully focused on its training dataset. We considered the best model, which had the lowest error and presumably the highest $R^{2}$.

After this, the activation function was considered, and three different alternatives were employed, as can be seen in Table 5. The best model proved to be ReLU.

Based on the results, the ReLU function seemed to be the best choice for the final model due to its low MAE and $R^{2}$. A similar study was conducted on the batch size, which is presented in Table 6.

The results showed that 32 was the best value for the batch size, so this was selected for the final model. Finally, the number of epochs was investigated to find the best result for this parameter, as shown in Table 7.

Clearly, the selected parameter for the number of epochs was 25,000, because this had the lowest error and the highest $R^{2}$. After careful the examination and setting of the right hyperparameters, the final model is presented in Table 8.

### 5.1. ANN Model for Energy Efficiency

The results of the model for energy efficiency are presented in Figure 12. This figure presents a comparative study of the predicted and analytical results. It is clear that the model captured the problem’s physics and understood the variations in the input parameters. The evaluation of the model was carried out using the mean absolute error and R_squared [49,50,51,52,53]. The model was able to achieve an MAE of 1.98%, and its $R^{2}$ was 0.98. Therefore, the model was very accurate, which explains the wide usage of these methods in different research.

### 5.2. ANN Model for Exergy Efficiency

Figure 13 shows the results of the predictions for the exergy efficiency. The results proved to be more accurate. The MAE of this model was 0.94%, and the $R^{2}$ was 1. The HL structure was 32,64,128,128,64,32.

### 5.3. ANN Model for Voltage

The predictions for the voltage are presented in Figure 14. The predictive model was based on the HL structure of 32,64,128,256,128,64,32. The model was able to achieve an MAE of 1.98%, and the $R^{2}$ was 0.98. The model was able to understand the physics behind the system and could predict the testing data satisfactorily.

### 5.4. ANN Model for $H_{2}$ Production

The hydrogen gas production was the final outcome of the studied process, so being able to accurately predict this parameter value could save a lot of time for analytical modeling. Therefore, the model in Figure 15 was employed for this purpose. The model had an MAE of 1.14% and an $R^{2}$ of 0.99. This showed how accurate the model was, and that it could be used instead of the formulations.

## 6. Conclusions

This article described the thermodynamic analysis of a polymer membrane electrolyzer connected to a solar tower’s concentrated solar system. The cell electrolyzer was modeled using electrochemical-mechanical equations, and the radiation intensity, current density, and electrolyzer parameters were investigated to determine how much hydrogen was produced and how well the electrolyzer performed. As the temperature increased and the pressure decreased, the electrolyzer’s performance increased. Due to the chemical reaction’s acceleration and the current exchange’s density, an increase in temperature increased the efficiency and exergy of the electrolyzer. As a result, the reduction in the activation and ohmic voltage dropped. Considering different Nafion membranes, it was found that Nafion 115 performed better than the other membranes due to its lower thickness and the subsequent decrease in the ohmic voltage drop. With an increase in the intensity of solar radiation from 125 $W/m^{2}$ to 320 $W/m^{2}$, the total energy efficiencies decreased by 14%, and the amount of hydrogen produced increased by 111%. Increasing the solar radiation in the same range reduced the energy and exergy efficiencies by 13.7% and 12.6%, respectively. Finally, the data from the research were gathered and employed to propose predictive models using a machine learning algorithm. An artificial neural network was utilized for the prediction of energy efficiency, exergy efficiency, voltage, and $H_{2}$ production. Hyperparameter tuning was used for setting the contributing parameters of the models. The results of the models proved to be very accurate, and, in general, they had an MAE lower than 1.98% and an $R^{2}$ higher than 0.98.

## Figures and Tables

**Figure 1 molecules-28-02649-f001:**
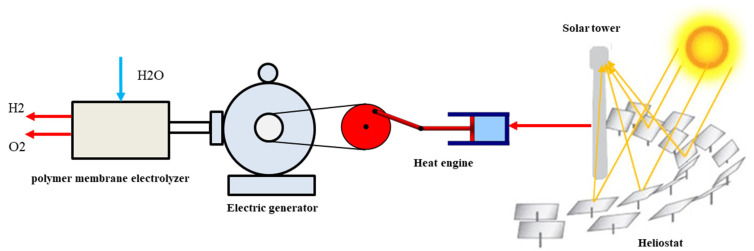
Schematic view of the system investigated in this article.

**Figure 2 molecules-28-02649-f002:**
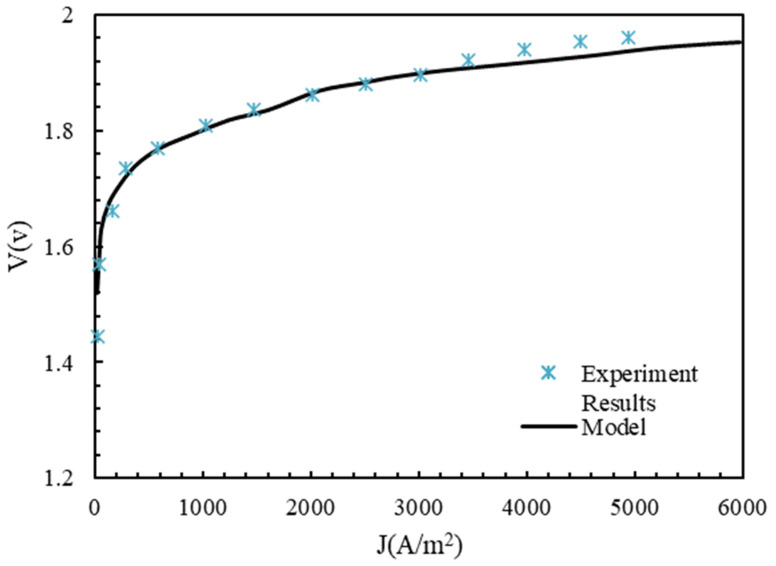
Comparison between experimental results of [43] and modeling results of this article.

**Figure 3 molecules-28-02649-f003:**
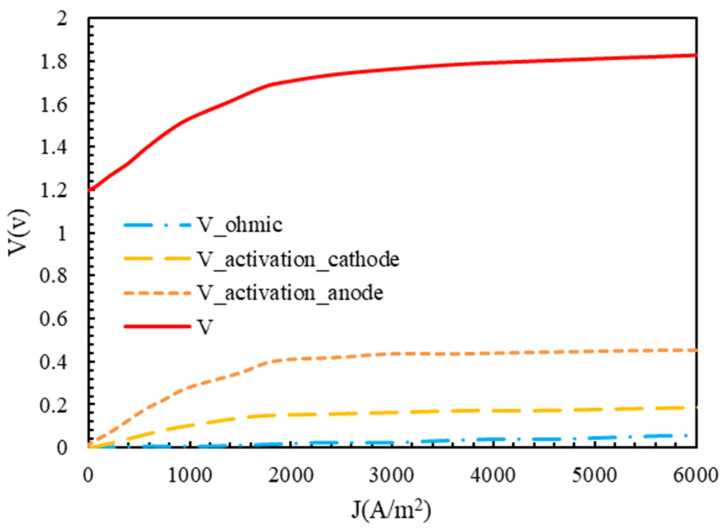
Polarization of polymer membrane electrolyzer.

**Figure 4 molecules-28-02649-f004:**
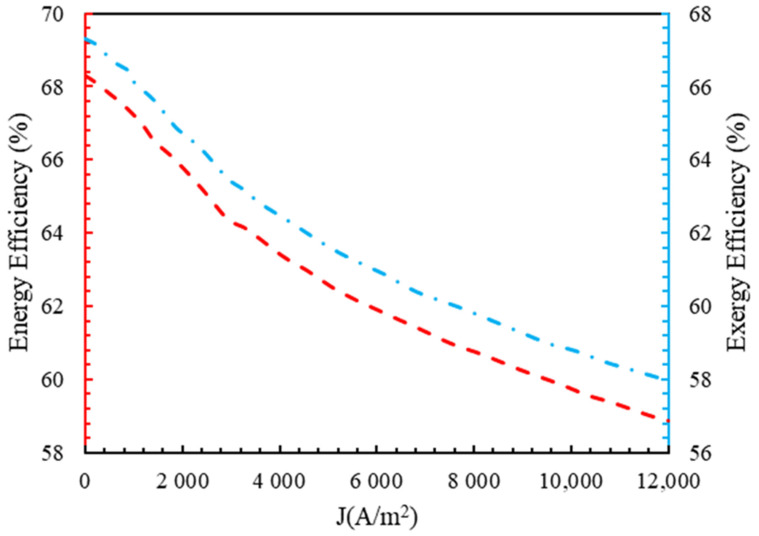
The energy and exergy efficiency of the polymer membrane electrolyzer in terms of current density.

**Figure 5 molecules-28-02649-f005:**
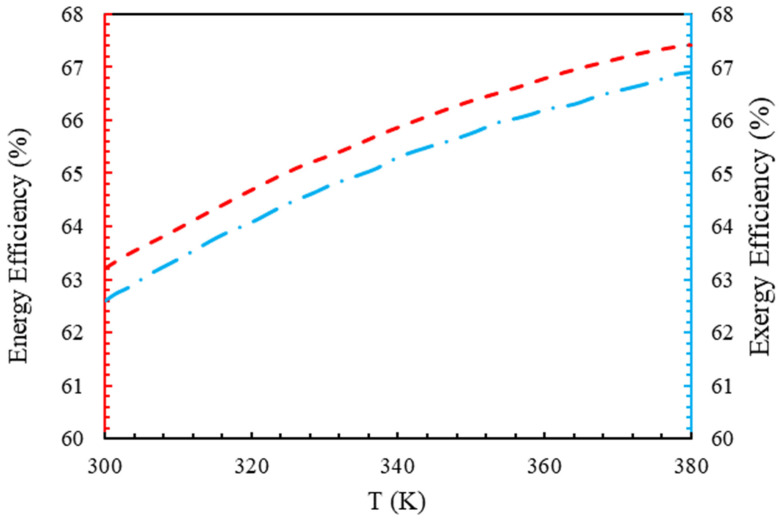
The effect of temperature on the energy and exergy efficiencies at a current density of 5000 $A/m^{2}$.

**Figure 6 molecules-28-02649-f006:**
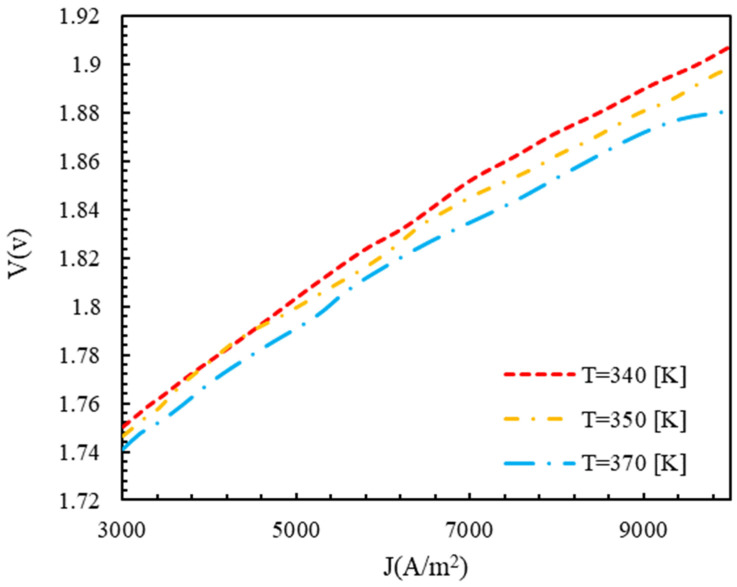
Polarization curve of polymer membrane electrolyzer at different temperatures with P=1 bar, electrode catalyst of platinum, and membrane of Nafion.

**Figure 7 molecules-28-02649-f007:**
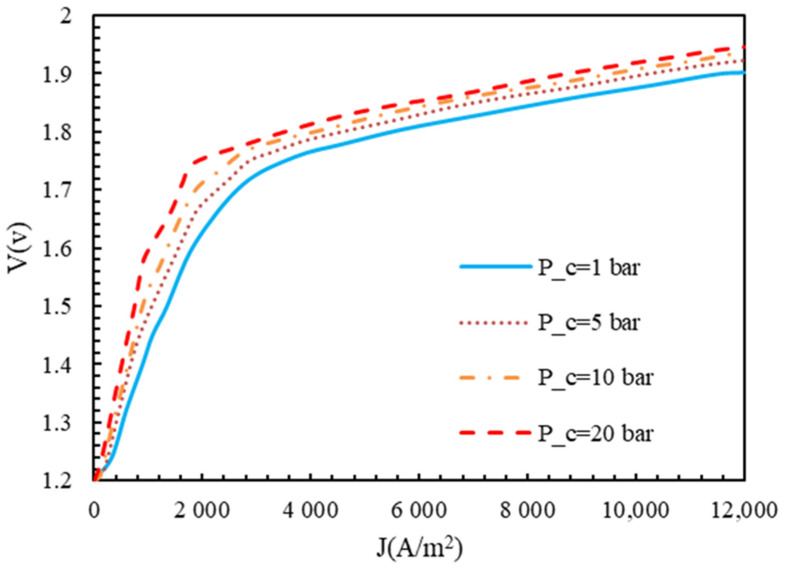
Polarization curve of polymer membrane electrolyzer at different pressures with T=350 K, electrode catalyst of platinum, and membrane of Nafion.

**Figure 8 molecules-28-02649-f008:**
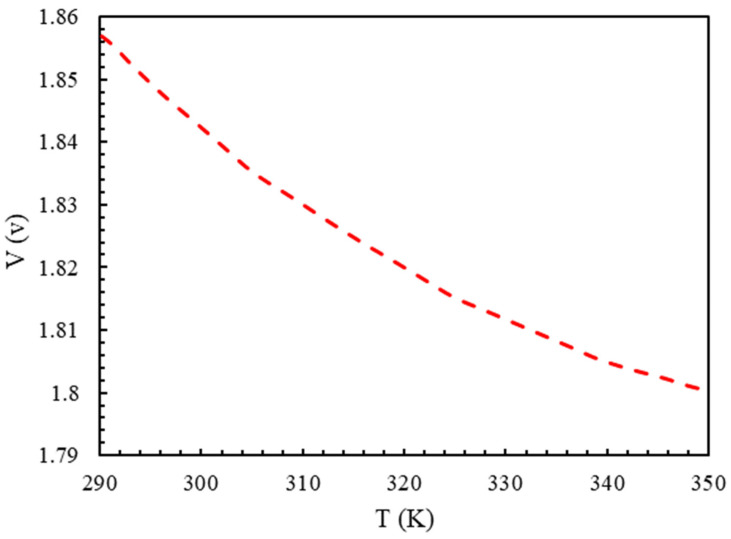
The effect of temperature on the electrolyzer’s cell voltage with a current density of 5000 $A/m^{2}$, P=1 bar, electrode catalyst of platinum, and membrane of Nafion.

**Figure 9 molecules-28-02649-f009:**
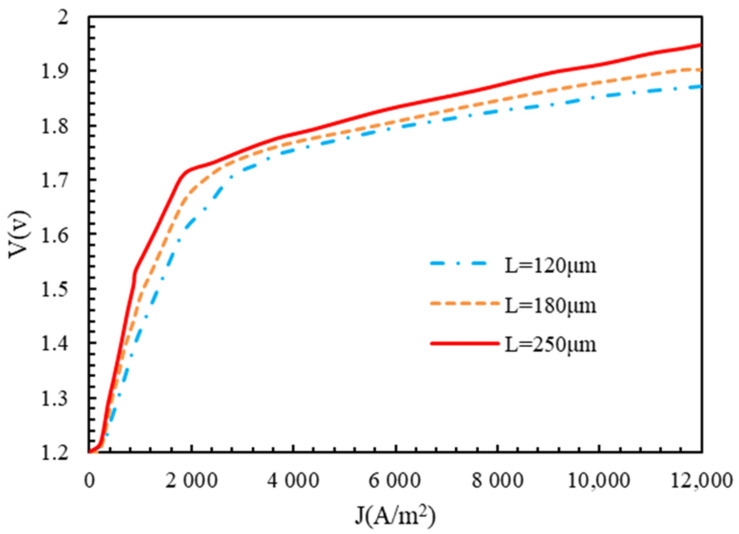
Polarization of polymer membrane electrolyzer for different membrane thicknesses with P=1 bar, T=350 K, electrode catalyst of platinum, and membrane of Nafion.

**Figure 10 molecules-28-02649-f010:**
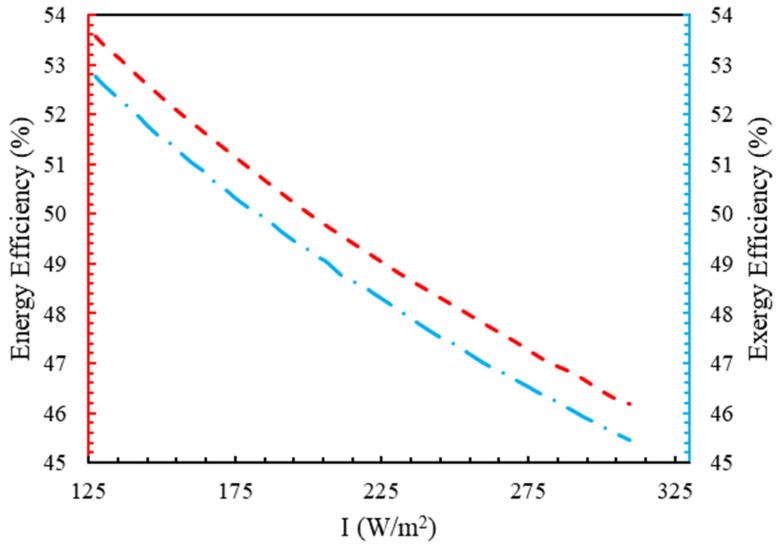
The trend of energy and exergy in terms of radiation intensity.

**Figure 11 molecules-28-02649-f011:**
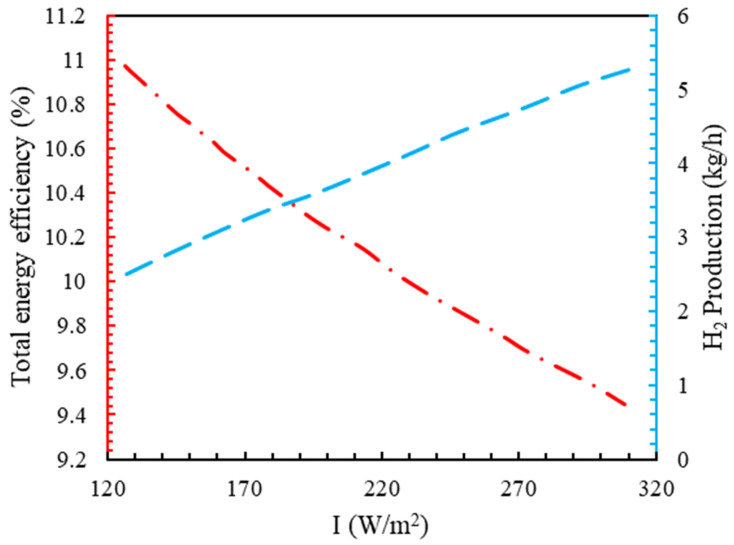
The effect of solar radiation on hydrogen production and overall energy efficiency.

**Figure 12 molecules-28-02649-f012:**
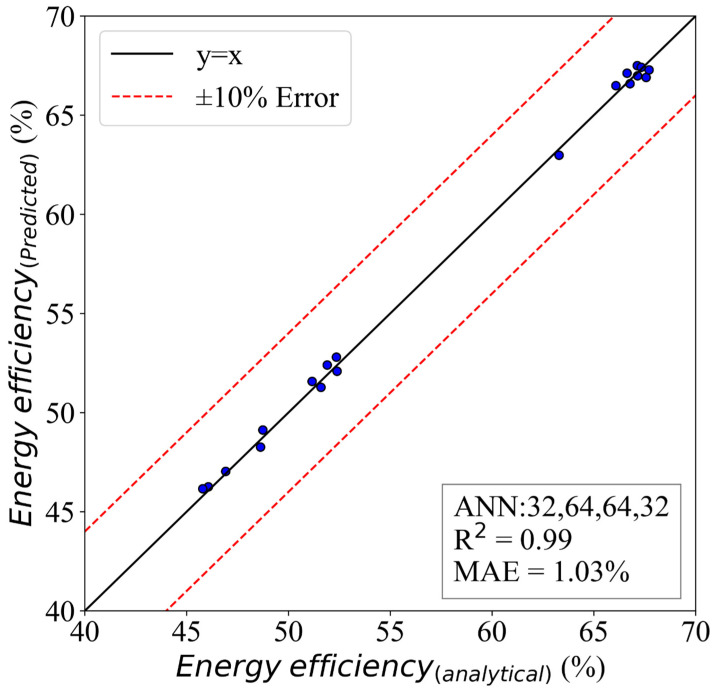
The ANN model of energy efficiency with an HL structure of (32,64,64,32).

**Figure 13 molecules-28-02649-f013:**
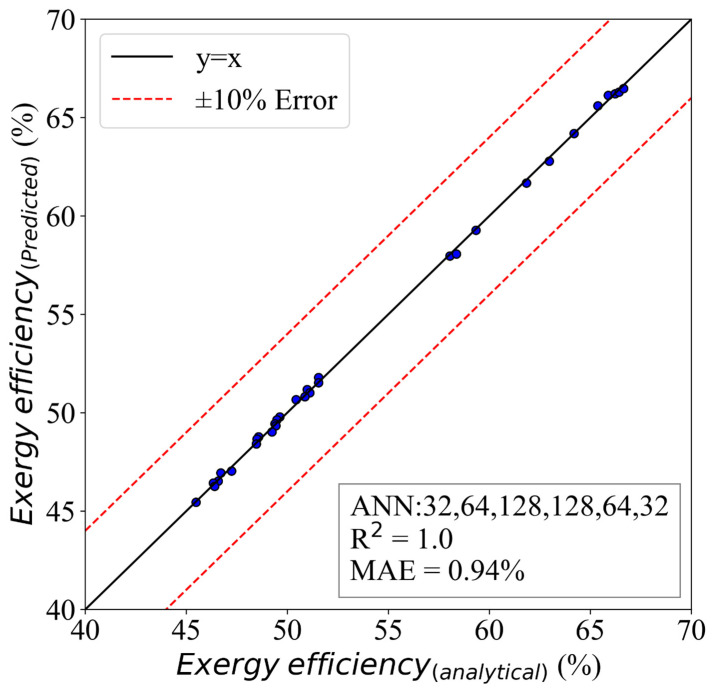
The ANN model of exergy efficiency with an HL structure of (32,64,128,128,64,32).

**Figure 14 molecules-28-02649-f014:**
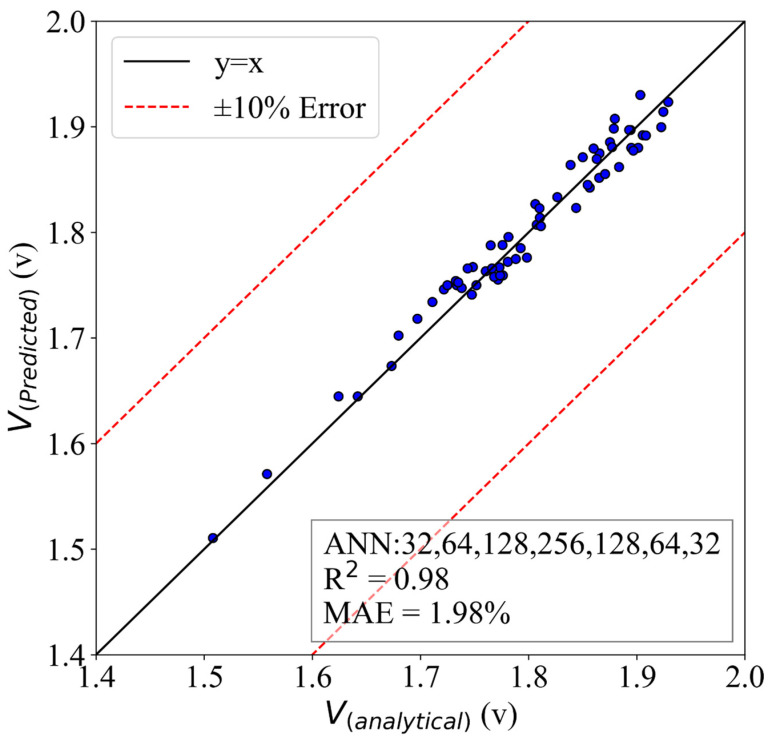
The ANN model of voltage with an HL structure of (32,64,128,256,128,64,32).

**Figure 15 molecules-28-02649-f015:**
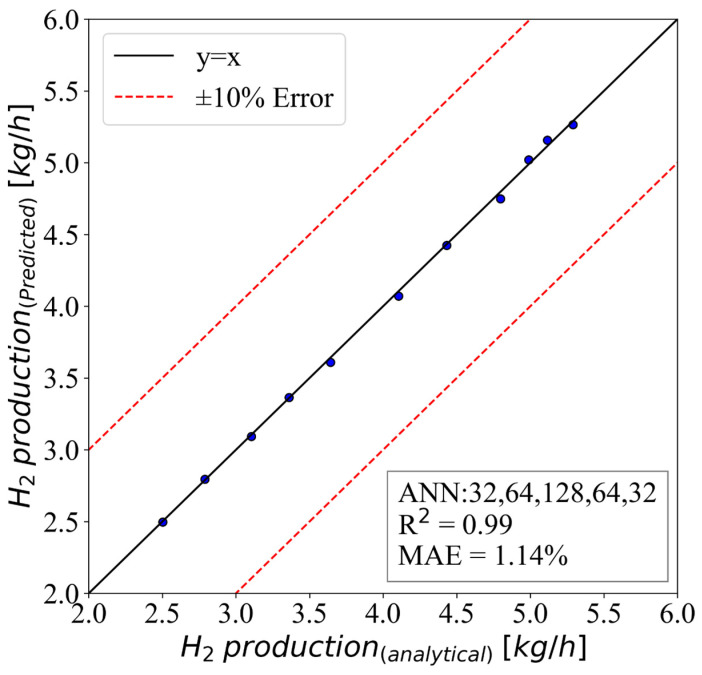
The ANN model of $H_{2}$ production with an HL structure of (32,64,128,64,32).

**Table 1 molecules-28-02649-t001:** Important parameters used to model the solar tower.

Solar Tower Parameter	Value
Heliostat efficiency	70%
Receiver’s efficiency	88%
Heliostat area	6100 $m^{2}$

**Table 2 molecules-28-02649-t002:** Fixed parameters for modeling the polymer membrane electrolyzer [37].

Parameter	Value
Water pressure ($P_{H_{2}O}$)	1 bar
Hydrogen pressure ($P_{H_{2}}$)	1 bar
Oxygen pressure ($P_{O_{2}}$)	1 bar
Membrane thickness (L)	183 μm
Electrolyzer temperature (T)	353 K
Lower heating value of hydrogen ($LHV_{H_{2}}$)	224 kJ/mol
The cathode activation energy ($E_{act,c}$)	18 kJ/mol
The anode activation energy ($E_{act,a}$)	76 kJ/mol

**Table 3 molecules-28-02649-t003:** Electrolyzer working conditions for validation.

Parameter	Value
Hydrogen pressure ($P_{H_{2}}$)	1 bar
Oxygen pressure ($P_{O_{2}}$)	1 bar
Water pressure ($P_{H_{2}O}$)	1 bar
Electrolyzer temperature (T)	353 K
Membrane thickness (L)	50 μm

**Table 4 molecules-28-02649-t004:** Hyperparameter tuning for the layer structure.

Model	Layer Structure	Mean Absolute Error (%)	$R^{2}$
1	(32)	3.14%	0.96
2	(32,64)	2.98%	0.97
3	(32,64,32)	2.86%	0.97
4	(32,64,64,32)	2.61%	0.98
5	(32,64,128,64,32)	2.43%	0.98
6	(32,64,128,128,64,32)	2.21%	0.98
7 *	(32,64,128,256,128,64,32)	2.05%	0.98
8	(32,64,128,256,256,128,64,32)	2.09%	0.98
9	(32,64,128,256,512,256,128,64,32)	2.13%	0.98

* is the selected model.

**Table 5 molecules-28-02649-t005:** Hyperparameter tuning for the activation function.

Model	Activation Function	Mean Absolute Error (%)	$R^{2}$
1	Linear	2.87%	0.97
2 *	ReLU	2.05%	0.98
3	Sigmoid	2.46%	0.98

* is the selected model.

**Table 6 molecules-28-02649-t006:** Hyperparameter tuning for batch size.

Model	Batch Size	Mean Absolute Error (%)	$R^{2}$
1	2	2.98%	0.97
2	4	2.47%	0.98
3	8	2.36%	0.98
4	16	2.39%	0.98
5 *	32	2.05%	0.98
6	64	2.86%	0.97

* is the selected model.

**Table 7 molecules-28-02649-t007:** Hyperparameter tuning for epochs.

Model	Epochs	Mean Absolute Error (%)	$R^{2}$
1	1500	5.21%	0.89
2	2500	4.76%	0.92
3	6000	3.15%	0.96
4	15,000	2.55%	0.98
5 *	25,000	1.98%	0.98
6	35,000	2.05%	0.98
7	45,000	2.08%	0.98
8	55,000	2.09%	0.98

* is the selected model.

**Table 8 molecules-28-02649-t008:** The ANN settings for the final model.

Hyperparameter	Energy Efficiency	Exergy Efficiency	Voltage	$H_{2}$ Production
Layer structure	(32,64,64,32)	(32,64,128,128,64,32)	(32,64,128,256,128,64,32)	(32,64,128,64,32)
Batch size	32	8	32	16
Epochs	45,000	35,000	25,000	35,000
Activation function	ReLU	Sigmoid	ReLU	ReLU

## Data Availability

The data presented in this study are available on request from the corresponding author.

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
