# Peer review of "Thermodynamics Investigation and Artificial Neural Network Prediction of Energy, Exergy, and Hydrogen Production from a Solar Thermochemical Plant Using a Polymer Membrane Electrolyzer"

_molecules, 2023, doi:10.3390/molecules28062649_

Round 1

Reviewer 1 Report

The authors of the paper “Thermodynamics investigation and artificial neural network 2 prediction of energy, exergy, and hydrogen production of a so-3 lar thermochemical plant of polymer membrane electrolyzer” report the hydrogen production in a solar installation, and using an electrolyzer. This thermodynamic research is interesting, but the authors must clarify different issues before it is published.

In the thermodynamic analysis section, the authors should define the parameters in the different equations (equation 1 to 13). What does Ah or I mean in equation 1? (for example). Equation 7 has a jump that should be fixed. There is terminology that has two ways of writing, as mechanically works in equations 10 and 12.

There are typical values that are not supported by a reference, for example, losses in the electric generator 50 to 62% (line 17). Carnot's yield is considered around 30 to 50% (line 169), but it is taken as a value for this study of 45%. Why? What motive allows you to take that value and not another? The authors should argue these values and then justify the subtractions obtained in this study.

 Other minor issues to consider:

In lines 48 and 49 they refer to renewable energies and give nuclear energy as an example. This type of energy can be very useful (without polemicizing), but it is not a renewable energy.

What is EES? (line 148).

The conductivity of membranes is protonic, not ionic (line 232).

In line 278 it talks of a current density of 350 K. Is there an error here?

Author Response

Response to reviewer #1

Reviewer's suggestions: The authors of the paper “Thermodynamics investigation and artificial neural network prediction of energy, exergy, and hydrogen production of a solar thermochemical plant of polymer membrane electrolyzer” report the hydrogen production in a solar installation, and using an electrolyzer. This thermodynamic research is interesting, but the authors must clarify different issues before it is published.

Comment 1: In the thermodynamic analysis section, the authors should define the parameters in the different equations (equation 1 to 13). What does Ah or I mean in equation 1? (for example). Equation 7 has a jump that should be fixed. There is terminology that has two ways of writing, as mechanically works in equations 10 and 12.

Authors' response to comments 1: Thank you for your fruitful comment. Per your kind comment all the mentioned parameters are defined in the manuscript, and the problem with equation 7 is fixed.

Comment 2: There are typical values that are not supported by a reference, for example, losses in the electric generator 50 to 62% (line 17). Carnot's yield is considered around 30 to 50% (line 169), but it is taken as a value for this study of 45%. Why? What motive allows you to take that value and not another? The authors should argue these values and then justify the subtractions obtained in this study.

Authors' response to comments 2: Thank you for your comment. The mentioned cases are assigned to their references. The case about the Carnot cycle is also referred to the source research, and based on the results of research most of the current conventional heat engines possess the efficiency of 45%.

Comment 3: In lines 48 and 49 they refer to renewable energies and give nuclear energy as an example. This type of energy can be very useful (without polemicizing), but it is not a renewable energy.

Authors' response to comments 3: Thank you for your comment. The nuclear energy is eliminated from the examples.

Comment 4: What is EES? (line 148).

Authors' response to comments 4: Thank you for your wise comment. We had used EES without mentioning the full name. Engineering Equation Solver (EES), is added to the manuscript.

Comment 5: The conductivity of membranes is protonic, not ionic (line 232).

Authors' response to comments 5: Thank you for your wise comment. Actually, the ionic conductivity seems to be correct, and also, in similar cases, the same terminology is utilized in the following research:

https://doi.org/10.1016/j.enconman.2008.03.018,

https://doi.org/10.1016/j.ijhydene.2010.11.122,

https://doi.org/10.1016/j.ijhydene.2017.09.126,

https://doi.org/10.1016/j.energy.2015.01.013

Comment 6: In line 278 it talks of a current density of 350 K. Is there an error here?

Authors' response to comments 6: Thank you for your wise comment. That is an error in the composition of the manuscript and it is revised to the following form:

Figure 4 shows the energy efficiency and exergy of the polymer membrane electrolyzer at 350 K.

Author Response

Reviewer's suggestions:

The authors present a model to predict the behavior of a polymer membrane electrolyzer in combination with a solar concentrating system. I think that the work is interesting and suitable for publication in Molecules, but it should be improved.

Comment 1: In the Thermodynamics analysis Section, all the used variables mut be defined.

Authors' response to comments 1: Thank you for your wise comment. Per your kind comment, all the variables are defined in the mentioned section.

Comment 2: Eqs. (4) and (6) must be justified and/or a reference would be given.

Authors' response to comments 2: Thank you for your comment. The mentioned equations are assigned to references.

Comment 3: The lower heating value (LHV)of hydrogen must be used when vapour water is involved in the reaction of water formation. As, according to data in Table2, the operating conditions are 1bar and 80ºC, and liquid water would be expected in this conditions, please, explain why the LHV has been used.

Authors' response to comments 3: Thank you for your wise comment. As Table 2 is referred to [37], the utilized value of LHV is mainly based on the results of this research. The controversy in using LHV and HHV in electrolysis is also mentioned on page 8 of:

Harrison, K.W., Remick, R., Hoskin, A. and Martin, G.D., 2010. Hydrogen production: fundamentals and case study summaries (No. NREL/CP-550-47302). National Renewable Energy Lab.(NREL), Golden, CO (United States).

Since still in many industries LHV is considered to be used instead of HHV even in this state the employed parameter seems to be correct. Also, the results of the modeling also proved to fully concur with the experimental results. Additionally, the same analysis is incorporated in the following research:

Ni, M., Leung, M.K. and Leung, D.Y., 2008. Energy and exergy analysis of hydrogen production by a proton exchange membrane (PEM) electrolyzer plant. Energy conversion and management, 49(10), pp.2748-2756.

Comment 4: All parameters in Eqs. (18) an (19) must be defined in the text.

Authors' response to comments 4: Thank you for your kind comment. The following is added to the manuscript:

Where, R is the universal gas constant, T is the temperature, and F is Faraday constant. Also, J is the current density.  is the activation energy.

Comment 5: Eq. (21) is an empirical expression obtained for Nafion membranes. It is possible that other kind of membranes had different values of the numerical parameter. Units for the membrane conductivity in Eq. (21) must be indicated.

Authors' response to comments 5: Thank you for your wise comment. Based on the source research of the mentioned equation, the only difference should be with , and based on the manuscript the selected value in the present study is fully validated with the experimental results. Also, the unit of the parameters are added to the manuscript.

Comment 6: Give a reference for Eqs. (24) and (26) or indicate that they correspond to the Faraday law.

Authors' response to comments 6: Thank you for your wise comment. The equations are assigned to their respective references.

Comment 7: The membrane thickness for modelling the polymer membrane electrolyzer (Table 2) is 183 micros, while the results used to validate the model correspond to a membrane of 50 microns, according to the data shown in Table 3. Please, clarify this point, as the thickness of the membrane is a key parameter.

Authors' response to comments 7: Thank you for your comment. Similar to the study of [37] in the manuscript, the validation is proposed to make sure the modeling is in the right direction and has the required accuracy. This, however, does not necessarily mean to carry out the rest of the modeling with those parameters. We agree that thickness plays an important role in the membrane performance, but we do not compare the results of different thickness at any point in the paper. As previously mentioned, the same is done in [37], where the validation is with the study of [42], but the thickness of the membrane differs with that case.

Comment 8: In pag. 9, line 278, I suppose that it should read temperature instead of current density.

Authors' response to comments 8: Thank you for your wise comment. The error is edited and the following is replaced:

Figure 4 shows the energy efficiency and exergy of the polymer membrane electrolyzer at 350 K.

Comment 9: In pag. 11, lines 299-302. Figure 5 show efficiencies and not voltages. Please, explain more extensively the claims about the influence of temperature on voltages.

Authors' response to comments 9: Thank you for your wise comment. The explanations are completed by adding the equations to which they refer. The manuscript is revised to the following:

A higher temperature means that more reactions are happening in the electrodes, leading to a more current exchange density, which reduces the activation voltage drop, based on equation 20. Electrolyzer cell voltage decreases as temperature increases, based on equations 20 and 21, therefore, electrical input is decreased [37].

Comment 10: Captions of Figures 6 and 7. “Polarization curve of….”

Authors' response to comments 10: Thank you for your wise comment. The captions are revised per your kind comment.

Comment 11: The value of the fixed operating parameters must be indicated in caption of Figures 6-9.

Authors' response to comments 11: Thank you for your wise comment. The fixed parameters are added to the captions as follows:

Figure 6. Polarization curve of polymer membrane electrolyzer in different temperatures with , electrode catalyst of platinum, and membrane of Nafion

Figure 7. Polarization curve of polymer membrane electrolyzer in different pressures with , electrode catalyst of platinum, and membrane of Nafion

Figure 8. The effect of temperature on the electrolyzer’s cell voltage with the current density of 5000 , , electrode catalyst of platinum, and membrane of Nafion

Figure 9. Polarization of polymer membrane electrolyzer with different membrane thickness with , , electrode catalyst of platinum, and membrane of Nafion

Comment 12: In Conclusions. Please, check the conclusions about the influence of temperature and pressure in the elecrtrolyzer performance.

Authors' response to comments 12: Thank you for your wise comment. There was an error in the conclusion, which is addressed per your comment. The manuscript reads as:

As the temperature increases and pressure decreases, the electrolyzer's performance increases.

Comment 13: References 17, 30, and 33 are incomplete.

Authors' response to comments 13: Thank you for your wise comment. The references are revised to the following:

17. K. W. Harrison, E. Hernández-Pacheco, M. Mann, H. Salehfar, 2006. Semiempirical model for determining PEM electrolyzer stack characteristic. Journal of fuel cell science and technology, 3(2), pp. 220-223.

30. Chand, K., & Paladino, O. 2023. Recent developments of membranes and electrocatalysts for the hydrogen production by anion exchange membrane water electrolysers: A review. Arabian Journal of Chemistry, 16(2), 104451.

33. Bazarah, A., Majlan, E. H., Husaini, T., Zainoodin, A., Alshami, I., Goh, J., & Masdar, M. S. 2022. Factors influencing the performance and durability of polymer electrolyte membrane water electrolyzer: A review. International Journal of Hydrogen Energy, 47(85), 35976-35989.

Round 2

Reviewer 1 Report

The modifications made by the authors are sufficient to be published this paper.

Author Response

Thank you so much for your positive feedback. We truly appreciate your effort for taking the time to review our manuscript 

Reviewer 2 Report

 The work has been properly revised.  I have only two comments:

-Equation [21] is an empirical expression obtained by Springer at al. for the proton conductivity of Nafion as a function of the temperature and membrane water content. I suppose that in the works cited by the authors this same membrane was utilized. However, for membranes with different structures, it is possible that other expressions would be valid. As the model developed in this work a Nafion membrane is used, the expression is correct, that it must be indicated in the text that is expression correspond to a Nafion membrane to avoid confusion.

T.E.Springer, T.A. Zawodzinski, S. Gottesfeld, Polymer electrolyte fuel cell model, J. Electrochem.Soc.138(8) (1991)2334-2342

-Reference 13 is incomplete

Author Response

Authors' response to comments 1: Thank you for your fruitful comment. The following is added to the manuscript to avoid confusion:

The  is the ionic conductivity of the Nafion membrane in terms of ( ) [41].

Comment 2: Reference 13 is incomplete.

Authors' response to comments 2: Thank you for your comment. The mentioned reference is revised per your comment to the following:

[13] He, Y., Wang, F., Du, G., Pan, L., Wang, K., Gerhard, R., Plath, R., Rozga, P. and Trnka, P., 2022. Revisiting the thermal ageing on the metallised polypropylene film capacitor: from device to dielectric film. High Voltage. pp.1-10. https://doi.org/10.1049/hve2.12278.
